# Embedded Machine Learning System for Muscle Patterns Detection in a Patient with Shoulder Disarticulation

**DOI:** 10.3390/s24113264

**Published:** 2024-05-21

**Authors:** Erick Guzmán-Quezada, Claudia Mancilla-Jiménez, Fernanda Rosas-Agraz, Rebeca Romo-Vázquez, Hugo Vélez-Pérez

**Affiliations:** 1Departamento de Electromecánica, Universidad Autónoma de Guadalajara, Guadalajara 45129, Mexico; jessica.rosas@edu.uag.mx; 2Departamento de Ciencias Computacionales, Dirección de Posgrados, Campus Internacional, Universidad Autónoma de Guadalajara, Guadalajara 45129, Mexico; claudia.mancilla@edu.uag.mx; 3Departamento de Biongeniería Traslacional, Centro Universitario de Ciencias Exactas e Ingenierías, Universidad de Guadalajara, Guadalajara 44430, Mexico; rebeca.romo@academicos.udg.mx (R.R.-V.); hugo.velez@academicos.udg.mx (H.V.-P.)

**Keywords:** artificial intelligence, electromyographic signals, prosthetic control systems, portable system, shoulder joint movements, Edge Impulse platform

## Abstract

The integration of artificial intelligence (AI) models in the classification of electromyographic (EMG) signals represents a significant advancement in the design of control systems for prostheses. This study explores the development of a portable system that classifies the electrical activity of three shoulder muscles in real time for actuator control, marking a milestone in the autonomy of prosthetic devices. Utilizing low-power microcontrollers, the system ensures continuous EMG signal recording, enhancing user mobility. Focusing on a case study—a 42-year-old man with left shoulder disarticulation—EMG activity was recorded over two days using a specifically designed electronic board. Data processing was performed using the Edge Impulse platform, renowned for its effectiveness in implementing AI on edge devices. The first day was dedicated to a training session with 150 repetitions spread across 30 trials and three different movements. Based on these data, the second day tested the AI model’s ability to classify EMG signals in new movement executions in real time. The results demonstrate the potential of portable AI-based systems for prosthetic control, offering accurate and swift EMG signal classification that enhances prosthetic user functionality and experience. This study not only underscores the feasibility of real-time EMG signal classification but also paves the way for future research on practical applications and improvements in the quality of life for prosthetic users.

## 1. Introduction

It has been estimated that approximately 10% of individuals with limb amputations are equipped with prosthetic devices. However, regrettably, only 7% of this population possesses the necessary knowledge or has received proper training to effectively utilize these devices [1]. The proper implantation of a prosthesis is essential during the rehabilitation process following limb amputation. When performed accurately, it enables individuals to fully reintegrate into their work and daily activities [2].

This, in particular, is critical when upper limb amputations are performed, as the hands and arms are mostly used in everyday life [3]. And also, it is well known that an amputation above the wrist leaves the limb with minimal functionality, which is considered a significant disability [4,5], especially in young patients [6].

The process of adjusting to an upper limb prosthesis presents a significant challenge for individuals in this patient population. Studies have reported a wide range of percentages for amputees’ adaptation to long-term prosthesis usage, ranging from 39% to 81% [7,8]. Despite the limited literature available on the factors influencing these outcomes in upper limb amputees [9,10,11], some studies suggest that this wide variation is closely associated with the level of satisfaction derived from prosthetic device utilization [12]. Therefore, greater attention should be given to improving rehabilitation strategies and refining prosthetic control systems to optimize outcomes for individuals with upper limb amputations [13].

The literature indicates that less than 9% of upper limb prosthesis users use them for daily activities where the use of both limbs is necessary. The majority of these individuals supplement this need with assistive devices [11], opting to reserve prostheses for specific activities. Furthermore, the importance of simplifying these devices and focusing on progressive adaptation is emphasized in order to avoid the rejection and psychological problems associated with adaptation to the prosthesis [10,14].

Among the individuals utilizing upper limb prostheses, those with shoulder disarticulation exhibit the highest rate of device abandonment, standing at 60%. They are followed by transhumeral amputees with a rate of 57%, and transradial users with a significantly lower rate of 6% [12]. Furthermore, these rates can vary depending on the type of prosthesis employed. Specifically, myoelectric prostheses exhibit abandonment rates of approximately 39%, passive hands at around 53%, and body-powered hooks at approximately 50% [15].

In recent years, there has been a notable growth in wireless and mobile technologies, leading to an increased utilization of communication protocols such as Bluetooth, Wi-Fi, Infrared, and others within the medical field. In the field of prosthetic research, there has been a notable surge in interest driven by the potential applications of advances in portable or mobile technologies. Specifically, electromyography (EMG) recording has experienced significant benefits through the transition from wired to wireless acquisition systems. The growing utilization of wireless EMG devices can be attributed to the necessity of validating electrophysiological measurements in diverse settings and situations [16,17,18], as well as their adaptability to various Human Device Interfaces (HDIs) [19,20,21] where they are expected to optimize usability, portability, and reliability in remote operational environments. Consequently, this enables the formulation of novel experimental protocols applicable in everyday contexts.

The EMG technique is employed to record the electrical activity generated by muscles as a signal, serving as the primary control method for myoelectric prosthetic systems. The underlying concept behind utilizing this signal is that amputees are able to generate phantom instructions. These instructions, although not resulting in limb activation, are continuously generated by the brain [22]. The majority of research efforts in this field primarily focus on investigating the flexion/extension of the forearm or the pronation/supination of the wrist, while the study of shoulder movements remains relatively limited. Conversely, other studies concentrate on exploring EMG characteristics that are utilized as input for artificial intelligence (AI) algorithms [23,24].

There are several EMG mobile devices that have achieved commercial success, including Myoware^®^ (Advancer Technologies, Raleigh, NC, USA, https://myoware.com/, accessed on 20 November 2023), SEN0240 (Meter Kit V2. DFRobot, Shanghai, China, https://www.dfrobot.com/product-1661.html, accessed on 21 November 2023), and others. These devices offer non-invasive and real-time acquisition of muscular activity. They offer ergonomic design, wireless functionality, and cost-effectiveness compared to other mobile EMG systems. However, it is important to note that these devices also have certain limitations, such as the fixed electrode location determined by the manufacturer’s design, constraints on electrode positioning, and restrictions on the number of channels employed. On the other hand, some studies have focused on the development of laboratory-level EMG prototypes [25,26,27,28,29], demonstrating the performance of their versions in local testing. However, these studies do not report the synchronized detection of multiple body muscles, which could significantly complicate the training of prosthetic devices. In addition, they do not report the application of these prototypes in patients who could be candidates for prosthetic use. They also lack a phase where EMG signals are analyzed and used for movement classification.

Edge Impulse (https://www.edgeimpulse.com, accessed on 15 Janaury 2024) is an emerging online platform specializing in the development of AI models designed specifically for embedded devices. The platform has successfully been utilized to create tools for recognizing hand movements in sign language, thereby facilitating the control of smart home devices. This implementation utilizes inertial sensors and has achieved an efficiency rate of 89.4% [30]. Additionally, Edge Impulse has developed applications in monitoring the operational condition of industrial equipment, enabling timely maintenance detection. In this context, the use of inertial sensors has resulted in an accuracy of 99.87% [31]. Another documented application involves the utilization of surveillance cameras to identify suspicious activities, triggering alarms in response to the detection of abnormal behavior [32].

Edge Impulse has also developed applications for discerning voice commands from databases comprising both young individuals and adults, with a reported accuracy of 97%, showcasing the potential to enhance interactions with embedded devices [33]. Additionally, this tool has successfully differentiated between distinct mosquito species, including those with potential lethality, solely based on the analysis of wing sounds. The conducted experiment reported a classification accuracy of 88% [34].

However, no instances of utilizing Edge Impulse for muscle activity recordings are identified in the existing literature. The closest analogous study focused on investigating the electrical activity produced by the heart, wherein a classifier was developed to enable the real-time prediction of normal or abnormal cardiac function. Remarkably, this research achieved an accuracy rate of 95% [35].

This research is focused on the analysis of the muscle patterns of a subject who suffered a shoulder disarticulation and is not currently a user of any prosthetic device. Muscular electrical activity was recorded using three EMG channels placed over the area proximal to the amputation stump. The structure of this paper is as follows: Section 2 describes the Methods followed to record the EMG activity of the volunteer for this experiment. The results obtained from the EMG building and comparison are presented in Section 3. Section 4 discusses the outcomes achieved in the development of the signal acquisition device and the application of AI, comparing these findings with those of other studies. Finally, Section 5 presents the conclusion of this research.

## 2. Materials and Methods

### 2.1. Participant

This study involved the enrollment of a male participant, aged 42, who experienced the loss of his left upper limb in a traumatic accident at the age of 21, resulting in a shoulder disarticulation; see Figure 1. The participant had no history of using any prosthetic devices. Prior to the commencement of the study, ethical approval (CEI/2021/001) was obtained from the Ethics Committee, and the volunteer provided informed consent, granting approval for his participation in the project. The investigation was conducted at the Laboratory of Electronics Specialized, within the Electromechanical Department of the Universidad Autónoma de Guadalajara (Mexico).

### 2.2. Data Acquisition

To record the muscular activity, Covidien-Kendall^TM^ (Medtronic, Minneapolis, MN, USA) brand Ag/AgCl foam electrodes, sized 2.54 cm in diameter, were placed on the pectoralis major, trapezius, and dorsal muscles on the left side of the participant.

The electrodes were connected to three electronic boards previously developed in the laboratory, based on the AD8232 integrated circuit designed by Analog Devices (Norwood, MA, USA) (Figure 2a). Although this component is used to monitor cardiac activity, it can also be used to measure other types of biosignals, including muscle activity. It also offers a low-power analog signal that a microcontroller’s analog-to-digital converter can read. The AD8232 has a two-pole high-pass filter and a three-pole low-pass filter, whose cutoff frequency can be modified by connecting externally passive components [36]. The 20 to 200 Hz frequency band was selected in this work, considering the ranges most frequently employed in the literature for EMG processing in prosthetic control (50 to 150 Hz) [37,38,39]. Also, a gain of ×1000 was considered, another parameter that can easily be adjusted with the equations proposed by the manufacturer.

In the calibration phase, a comparison was made with the Biopac^®^ system (BIOPAC Systems Inc., Goleta, CA, USA), specifically, the model BIOPAC STUDENT LAB BASIC SYSTEM MP36 (BIOPAC Systems Inc., Goleta, CA, USA). A total of six electrodes were placed: four of them in the chest area, where two were connected directly to the professional equipment and the other two to our prototype. The remaining two electrodes were used as reference, one for each acquisition system. The signals from our prototype were acquired using the Arduino platform, which was also used in later stages; see Figure 2b).

This comparison allowed us to validate our circuit and make the necessary adjustments before proceeding to the next stage of the experiment. Both signals were sampled at a rate of 2000 samples per second.

Finally, during the experimental stage, a Nano 33 BLE Sense Arduino (Arduino, Turin, Italy:) evaluation board was used to perform the digitizing process. Only three of eight analog-to-digital converters (ADCs) were included on this board, leaving the opportunity to increase the number of muscles recorded; see Figure 2c). A 10-bit resolution was considered for each ADC. Data were sampled at 1000 data per second and recorded using serial communication between the evaluation board and the Edge Impulse internet platform.

### 2.3. Experimental Test

For the experiment, the volunteer remained seated in front of a monitor with his spine resting on the back of the chair and at a reasonable distance from the screen. The EMG electrodes were then placed on the muscles of interest (pectoral, trapezius, and dorsal).

The experiment spanned two days. On the first day, the participant performed five series of ten arm abduction movements, ten arm adduction movements, and ten arm raises. All the movements were executed with the muscles affected by the shoulder disarticulation. At the end of each series, a 3 min pause for rest was also considered.

A visual interface developed in Python (ver 3.2) showed the participant the movement that would be executed, followed by a 1.5 s wait window, and finally, a 3 s window where the execution of the movement would take place was displayed. Each trial was followed by a 3 s rest pause before the next trial (Figure 3a,b). Data were sent through the interface to the Edge Impulse platform assigning a numerical marker to each event. Since these values were only used to identify the start, end, and type of each attempt, they did not affect the experiment.

To avoid the possibility of volunteers memorizing the trials, they were all selected randomly. A database consisting of 150 trial executions (50 per movement) was obtained at the end of the experiment and used to train an AI model.

Figure 3c shows our participant during the recording of muscle signals. In the table, the prototype consisting of electronic cards is observed, together with the electrodes that are connected over the muscles of interest.

Following the same exercises as the previous day, the model was loaded onto the evaluation circuit board (parameters will be described in the next section). The only difference was that the volunteer was free to decide when to perform a new movement since he did not receive instructions to start a new trial. Also, the AI model was running all the time, simulating the functioning in daily life. To provide feedback to our participant and help him improve his training, a virtual avatar received commands from the AI model and mimicked the volunteer’s movement (Figure 4).

### 2.4. Data Analysis

Edge Impulse is a tool for developing machine learning models that can be used on specific development electronic board platforms by recording data from any sensor. In this study, data from the three target muscles mentioned above were recorded at a sampling rate of 100 samples per second. Each trial was separated as an independent window and assigned a category corresponding to the movement performed. The final working windows were centered between Triggers 610 and 620, related to arm raise; between 710 and 720, associated with arm adduction; and finally, between 810 and 820, linked to arm abduction. Each of these windows lasted 3 s—the time during which the participant performed the movement. Furthermore, each window had 300 samples for each trial.

After the data were sorted, an 80/20 (120/30 trials) data ratio for training and validation was used in a 5-fold cross-validation testing process. Edge Impulse uses digital Butterworth filters to extract the frequency and power characteristics of each sample, reducing the frequency spectrum as much as possible for easier processing. Subsequently, a neural network designed using Python Keras (ver 3) is applied, which is the main engine of Edge Impulse for the development of artificial intelligence models. One hundred epochs were selected as training cycles and the number of characteristics varied according to the model evaluated. Two dense layers were maintained for all models, the first with twenty neurons and the second with ten neurons, all associated with the three classes that could be predicted, and all using a learning rate of 0.005 or 0.0005.

## 3. Results

### 3.1. Electronic Board Assembly

As Figure 5 shows, a multilayer copper-printed board was constructed, based on the design proposed in Section 2.2. All passive electronic components around the central integrated circuit were carefully calculated according to the specifications of the AD8232 datasheet [36]. These components interact with the internal operational amplifiers of the integrated circuit to provide a bandpass filter with cutoff frequencies at 23.41 and 224.5 Hz. These values are close to those suggested at the circuit design stage and are within the frequency range commonly studied in the literature [37,38,39]. In addition, the circuit detects the presence of the electrodes as soon as they are connected, and in the event of incorrect placement or absence of contact, the device goes into standby mode, allowing verification and energy saving.

Finally, the output signal of the circuit is transmitted in analog form to the Arduino microcontroller and, although it has already been amplified and filtered, it can be digitally post-processed if necessary. Figure 6 shows the average of the recorded EMG signals from all trials with each of the three muscle movements under study. The series of graphs exhibits the activity of three different muscles during the execution of a specific movement. Each graph corresponds to a different muscle, labeled from left to right as pectoral, trapezius, and dorsal. The ‘Left’ row, shown in blue, indicates that these data represent the shoulder adduction movement. In the middle row, the graphs in red are marked with the legend ‘Right’ and display the shoulder abduction movement. Lastly, the bottom row in green, under the legend ‘Up’, represents the elevation of the arm. Each graph plots the corresponding signal as a function of time, which extends along the horizontal axis marked as ‘Time (sec)’ in seconds.

### 3.2. Model Training

For feature extraction, three different methods were employed, two of which were already preconfigured in the Edge Impulse platform. The third method is a combination of both feature types. The first method was the Flatten method. This option focuses on extracting time characteristics, applying different statistics to a time window of the signal and assigning a value for each time window. These statistics include mean, maximum and minimum values, root mean square, standard deviation, skewness, and kurtosis. Two models were generated using this method. In the first, the learning rate was set at 0.0005, which resulted in a data prediction accuracy of 89.5%. In this case, the model confused certain movements of the left arm with those of the right arm. In contrast, the right and up trials were correctly identified by the predictor.

On the other hand, the learning rate for the second model was set to 0.005, ten times higher than that of the previous model. This value yielded a 100% accuracy for all three movements. Some precautions were taken in choosing learning rate values when designing both models. This is because small changes in values could lead to overtraining of the data (Figure 7).

The second method was focused on spectral analysis to obtain the spectral power of the signals and generate feature extraction. Again, two models were developed, keeping the same learning rates as the previous method. This method achieved 94.70% accuracy with a learning rate of 0.0005, a value higher than that of the previous method. In the same way, it also confused the left movements but this time with the up movements. Nevertheless, it was able to predict 90.9% of the cases, surpassing the 81.8% accuracy of the previous method. The rest of the movements were predicted without any problem. Increasing the learning rate also exhibited no difference, as it also achieved an accuracy of 94.7% and again confused left movements, which already had a predictive value of 90.9%, with respect to right movements (Figure 7).

Several runs of this model were performed to find the best parameters and obtain the best classifier. During these runs, parameters such as the learning rate, the number of layers of the neural network, and the number of neurons in each layer were adjusted, which improved the accuracy and generalization capacity of the model.

Although the best feature extraction model was the Flatten method with a learning rate of 0.005, a model integrating the previous two methods was proposed to compare the results. As with previous tests, the learning rate remained the same, and it was found that with a learning rate of 0.0005, an accuracy of 89.5% was achieved. This did not result in an increase over the Flatten method but a decrease of about 5% compared to the spectral analysis. Finally, 100% accuracy was achieved with a learning rate of 0.005. This result is the same as that of the Flatten method, but an increase of over 5% was observed for spectral analysis (Figure 7).

### 3.3. Model Testing

According to the previous results, only models with 100% accuracy were retained for the next phases of the tests and used to validate the test data previously separated. Thus, the Flatten model, with a learning rate of 0.005, achieved an accuracy of 95.24%. Again, movement to the left was the most difficult to classify, predicting only 85.7% of the data and identifying the rest as unknown (Figure 8).

Similarly, the Flatten + Spectral Analysis model, with a learning rate of 0.005, did not differ from the test data. It also achieved 95.24% accuracy and had problems predicting the left arm movements, confusing them with unknown movements.

These discrepancies in performance suggest that the models may struggle with certain types of motions, indicating a potential area for improvement. To improve performance, future work could include experimentation with different feature extraction methods, hyperparameter tuning, or the use of data augmentation techniques to better handle challenging motion patterns.

### 3.4. Online Classification

All the previous results correspond to data obtained on the first day of the experiment. As mentioned earlier, a second test was conducted the next day, involving the real-time detection of the three movements while the volunteer performed them freely.

For this second test, the electrodes were placed in the same anatomical positions as the day before. Later, the participant not only observed the interface used previously, where he waited for a new movement instruction but also observed an avatar (Figure 4) that mimicked the volunteer’s movements when the AI models predicted them correctly. To do this, both AI models, obtained through Edge Impulse, were previously loaded into the Arduino Nano 33 BLE Sense microcontroller. The volunteer was then asked to complete only two test sessions per model. The same number of trials per session was adhered to, and 60 new retrials were conducted for each model.

The left side of Figure 9 shows the results of the online Flatten model, which uses only the temporal characteristics of the signals to predict the new movements of the participant. This method achieved an accuracy of 95%, a result close to that obtained during the training stage. On the other hand, the classifier’s performance in predicting all the movements was correct, confusing left and right movements only a few times. Finally, the second model, which used the temporal and spectral characteristics of the signals, is shown at the right of Figure 9. In this case, the data prediction accuracy decreased slightly compared to the corresponding training model, going from 95.24% to 93.34%, which is a very insignificant difference.

The effectiveness of the online classification models in predicting muscle movements in real time aligns with the study’s objectives of enhancing prosthetic control. The practical implications of this are significant, as accurate real-time prediction enhances the usability of prosthetic devices, contributing to the autonomy of users. The slight decrease in accuracy observed during the real-time testing suggests that future models could benefit from further refinement to enhance robustness in real-world settings.

## 4. Discussion

In this research, we have presented a robust AI-enabled EMG classification system tailored for a shoulder disarticulation case. Our results demonstrate unprecedented real-time accuracy, particularly with the Flatten method at a learning rate of 0.005. These findings are supported by the data presented in Table 1, where we compare the performance of our EMG prototype with existing solutions. Unlike the other prototypes listed, which have a broader frequency range for EMG signal acquisition, our system operates effectively within a narrower band (23.41–224.5 Hz), optimizing signal clarity and processing efficiency. The data in Table 1 also illustrate the electrical characteristics of various devices and potential prototypes, aiding in the design of our electromyograph and helping us develop a competitive version.

Our portable EMG prototype, leveraging wireless and wired communication, stands out by balancing the sampling frequency with the precision of muscle activity readings. This balance is crucial, as Allard et al. [40] and Prakash et al. [41] demonstrated that a higher sampling frequency does not necessarily translate into better motion prediction accuracy. The slight drop in accuracy during online classification, compared to model training phases, is likely due to the dynamic nature of real-life movement execution as opposed to controlled experimental conditions. However, our system’s performance remains superior when benchmarked against other studies, which commonly report decreased model accuracy when transitioning from offline to online testing scenarios. The inclusion of a virtual avatar as biofeedback represents an innovative step in prosthetic control, as it may enhance user adaptability to prosthetic devices by providing visual cues aligned with detected EMG patterns. This approach could address the high abandonment rates observed in shoulder disarticulation prostheses, offering a more intuitive and user-friendly interaction with the device.

In terms of prototype size, our system is smaller than other systems in Table 1, a crucial advantage in prosthetic control where space and weight are critical factors. Allard et al. [40] mention a prototype size of 50 mm², but it is part of a bracelet integrating eight sensors, limiting its use to specific body parts and not suitable for all types of prosthetics as ours is.

The studies referenced in [25,26,27,28,29] focused on EMG signal acquisition but did not address synchronized detection of multiple body muscles, which is important for training prosthetic devices. Moreover, these studies did not apply their prototypes to potential prosthetic users or analyze EMG signals for movement classification, as we did. Our embedded system utilizes AI models, a novel approach not previously reported in the literature.

A limitation of this study was the small participant pool, which could impact the generalizability of the findings. Future studies should address these challenges by including more participants and refining the prototype for broader applications.

The generalizability of the findings is crucial for future research directions. The system’s effectiveness should be tested across different populations, types of amputations, and real-world settings to determine its broader applicability.

It is important to consider the ethical and societal implications of this research. Privacy concerns, user autonomy, and the impact of technological advancements on healthcare delivery are key factors. Addressing these aspects broadens the scope of the discussion and highlights the broader implications of the study beyond technical aspects.

The presented EMG prototype and AI methodology highlight the way for developing more accurate and user-friendly prosthetic control systems. Future studies should expand the participant pool and explore the methodology’s efficacy across different types of limb movements and amputations. By doing so, we can refine the system’s accuracy and make significant strides toward a better integration of prosthetic devices into the daily lives of amputees.

## 5. Conclusions

This study has successfully demonstrated the integration of artificial intelligence techniques into the design of a portable EMG-based prosthetic control system, showing promise in advancing the autonomy and functionality of prosthetic devices for individuals with upper limb amputations. Over the course of two days, our subject-specific approach allowed for the collection of robust data and the development of an embedded AI model capable of the real-time classification of shoulder muscle movements with impressive accuracy.

Our findings, as highlighted by the confusion matrices in Figure 6 and Figure 8, underscore the system’s proficiency in distinguishing between different directional movements, although recognizing left arm movements remains a challenge. This challenge is reflective of the complexity of muscular patterns and the subtleties inherent in accurate EMG signal interpretation.

The successful deployment of this system onto a low-power microcontroller platform, the Arduino Nano 33 BLE Sense, illustrates the feasibility of portable and real-time muscle pattern recognition. This accomplishment is a significant step toward the development of more adaptive and intuitive control systems for prosthetic users. Our research contributes to the burgeoning field of embedded machine learning, particularly within the context of biomedical applications. The consistency of our system’s performance from the training phase to real-world application provides a methodological framework that could potentially be adapted to other forms of prosthetic control and rehabilitation technologies.

As we look to the future, the scope for expanding this study is considerable. It will be essential to replicate this work with a broader participant base and to investigate the application of the technology to different types of limb movements and various levels of limb loss. It is our hope that these advances will not only further the capabilities of prosthetic devices but also tangibly improve the quality of life for their users by offering them greater control and independence.

The potential demonstrated by the integration of AI models, such as those provided by Edge Impulse, paves the way for innovative solutions that could transform the landscape of prosthetic design and user experience. Our study serves as a beacon for future explorations in this exciting and rapidly evolving field.

## Figures and Tables

**Figure 1 sensors-24-03264-f001:**
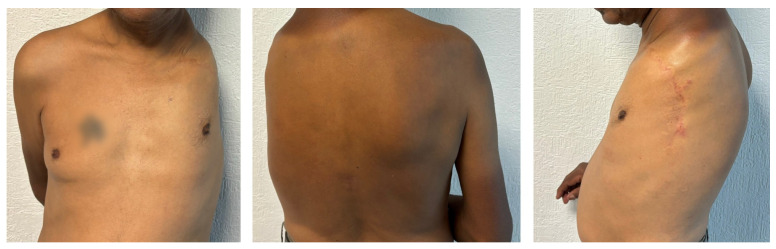
Front, back and side views of the participant. The scar resulting from the accident can be seen on the left side.

**Figure 2 sensors-24-03264-f002:**
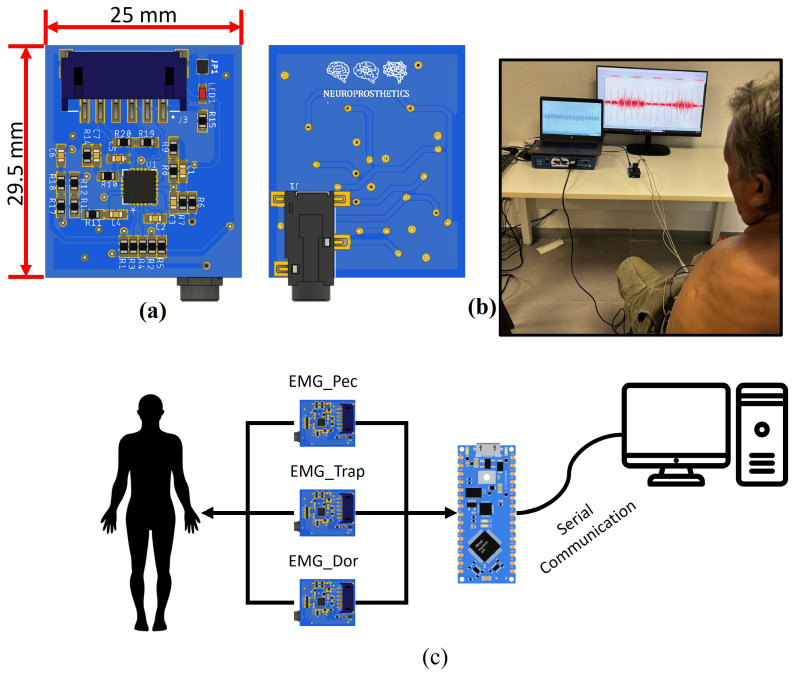
At the top left, the electronic board’s design and size. The top and bottom views are displayed in (**a**). At the top right, the patient during calibration, in (**b**). Bottom, the prototype for recording and transmitting EMG data from the participant to the computer (**c**).

**Figure 3 sensors-24-03264-f003:**
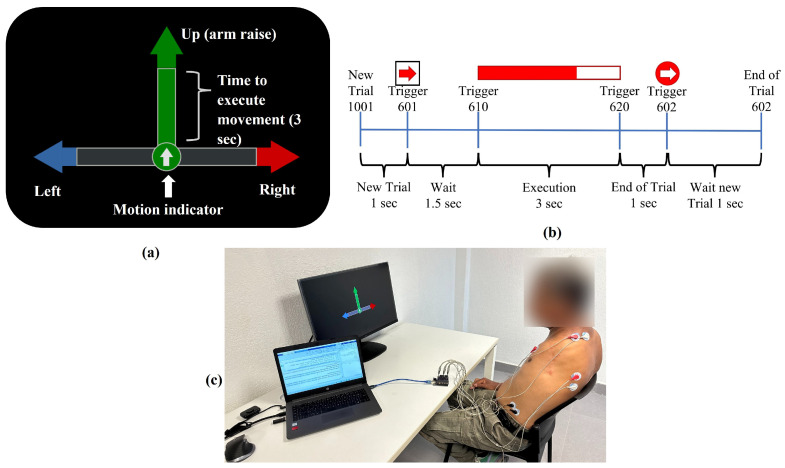
(**a**) Example of the interface developed in Python to indicate the sequence of each movement. (**b**) The temporal sequence followed for the execution of a movement of the participant’s left arm. All the movements have the same time intervals, the only difference being the values of the triggers. (**c**) Setup implemented during the experiment.

**Figure 4 sensors-24-03264-f004:**
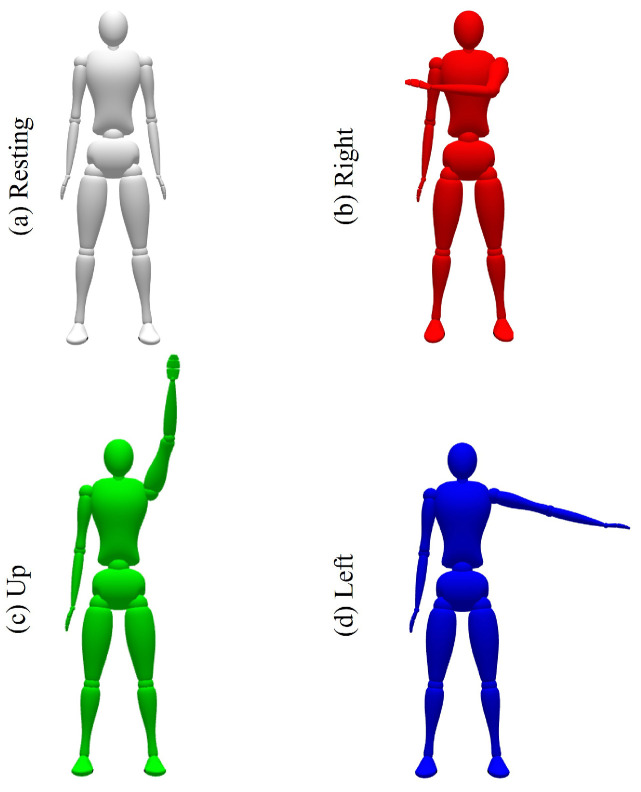
Virtual avatar used to provide feedback and mimic the volunteer’s intended movements, (**a**) white, resting activity, (**b**) red, right movement, (**c**) green, up movement, (**d**) blue, left movement.

**Figure 5 sensors-24-03264-f005:**
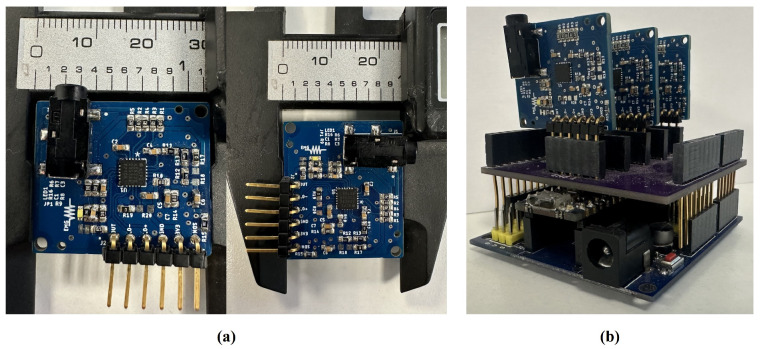
(**a**) Final construction of an EMG prototype based on the AD8232 integrated circuit. (**b**) The 3 EMG channels connected to the Arduino Nano 33 BLE Sense Arduino.

**Figure 6 sensors-24-03264-f006:**
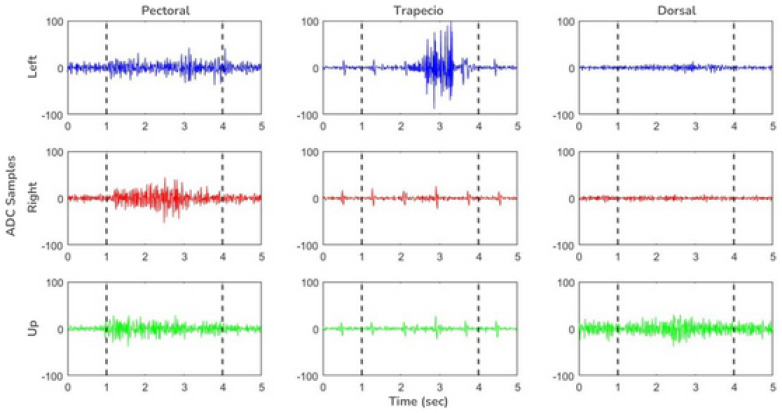
Average of the raw signals of the three muscle recording channels during the different movements. On the X-axis, the recording time for each sample (5 s). On the Y-axis, the values of the ADC. Dotted lines indicate the 3-s window during which features were extracted from the AI model.

**Figure 7 sensors-24-03264-f007:**
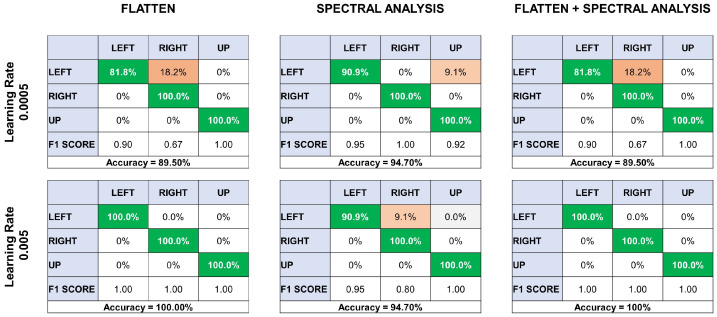
Confusion matrices showcasing classification accuracy for directional movements at learning rates of 0.0005 and 0.005, using data flattening, spectral analysis, and a combination of both. Results are depicted with F1 scores and overall model accuracy.

**Figure 8 sensors-24-03264-f008:**
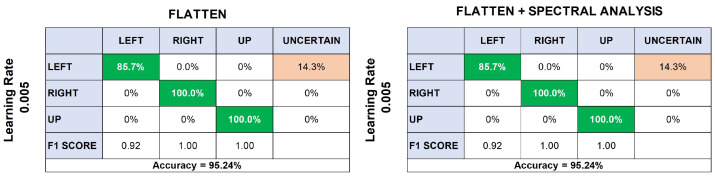
Comparison of classification performance using Flatten and Flatten + Spectral Analysis models at a learning rate of 0.005. Both models achieved an accuracy of 95.24%, with the left movement prediction being the most challenging, having an 85.7% success rate and the remainder classified as uncertain.

**Figure 9 sensors-24-03264-f009:**
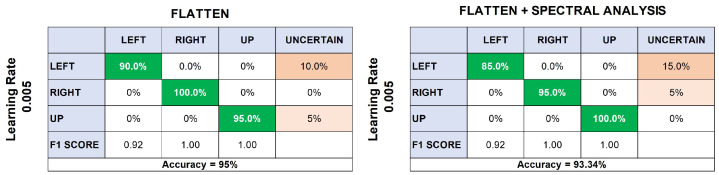
Results from the second day real-time movement detection tests using Flatten and Flatten + Spectral Analysis models. The Flatten model had 95% accuracy, and the combined model exhibited a minor drop to 93.34% accuracy, with some difficulty in predicting left movements.

**Table 1 sensors-24-03264-t001:** Electrical characteristics of commercial devices and prototypes focused on electromyography used in rehabilitation and prosthetic techniques at a research level.

Article	Allard [40].	Prakash [41].	Sattar [42].	Vavrinsky [43].	Walter [44].	Liu [45].	This EMG Prototype
Application	EMG band design, and hand recognition	Prosthetic control	Prosthetic control with Myo armband	EMG design	EMG design	EMG design, and rehabilitation.	Portable EMG design
Prototype/ Product	Prototype	Prototype	Product	Prototype	Prototype	Prototype	Prototype
Communication	Wireless	Wired	Wireless	Wireless	– –	Wireless	Wireless and wired
Num. of channels	10	1	8	– –	1	4	1 to 6
Sampling frequency [Hz]	1000	2000	200	– –	1000	1000	1000
Frequency range [Hz]	20–500	11.4–323.7	– –	2–300	10–500	20–500	20–200
Type of electrode	Electroless nickel immersion gold (ENIG)	Ag/AgCl	– –	PCB Electrode	Ag/AgCl	Ag/AgCl	Ag/AgCl
Weight [g]	62	42	– –	– –	25	– –	20
Size	50 mm^2^	25 × 70 mm	– –	– –	57 × 36 mm	35 × 25 mm	29.5 × 25 mm

## Data Availability

The experimental data can be found at the following GitHub link (accessed on 15 May 2024).

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
