# Peer review of "Embedded Machine Learning System for Muscle Patterns Detection in a Patient with Shoulder Disarticulation"

_sensors, 2024, doi:10.3390/s24113264_

Round 1

Reviewer 1 Report

Comments and Suggestions for Authors

A portable EMG system was implemented with AL model. Figure and Table are well organized. Minor English grammar issues. However, it is hard to recognize the novelty of the manuscript compared to previous portable EMG system. There are some suggestive comments.

1) The literature background based on previous EMG system was not described in Introduction. 

2) Authors need to correct pectoral, trapezius and dorsal to pectoral, trapezius, and dorsal.

3) In Line 136, "To prevent volunteers from". From what ?

4) No description of Figure 4 in the manuscript.

5) Authors need to emphasize the novely of the manuscript compared to previous portable EMG system. 

6) Abbreivated journal names in References need to be used.

7) There is only one person case so authors need to provide the similarity or comparison description if authors use another case. Otherwise, authors need to verify the open-data in the web.

8) In Figure 3, why another movement is not considered ? Ex. Left arm movement.

9) How to operate the EMG prototype wirelessly ? 

10) Author Contributions format is wrong.

Comments on the Quality of English Language

None

Reviewer 2 Report

Comments and Suggestions for Authors

Dear authors

Review for the article "Embedded Machine Learning System for Muscle Patterns Detection in a Patient with Shoulder Disarticulation"

1.      The introduction effectively outlines the context and significance of the research topic by highlighting the challenges faced by individuals with limb amputations, particularly concerning the utilization of prosthetic devices. The use of statistics to underscore the prevalence of prosthetic device abandonment among this population adds weight to the importance of the study. The introduction sets a clear direction by emphasizing the critical need for improved rehabilitation strategies and refined prosthetic control systems to enhance outcomes for individuals with upper limb amputations. By contextualizing the research within the broader challenges of prosthetic device usage and rehabilitation, the introduction effectively engages the reader and underscores the relevance of the study.

However, while the introduction provides a strong overview of the problem statement and its significance, it could benefit from further contextualization within the existing literature. While it briefly mentions the limited literature available on factors influencing outcomes for upper limb amputees, a more thorough review of relevant studies would provide a stronger foundation for the current research. Additionally, while the introduction sets the stage for the study, it could be strengthened by explicitly stating the research objectives or hypotheses. Clearly articulating the specific aims of the study would help guide the reader and provide a more focused framework for evaluating the research outcomes.

2.      The Materials and Methods section of the article "Embedded Machine Learning System for Muscle Patterns Detection in a Patient with Shoulder Disarticulation" provides a detailed description of the participant, data acquisition procedures, experimental tests, and data analysis techniques employed in the study.

2.1. Participant: The inclusion of a male participant aged 42 who experienced shoulder disarticulation due to a traumatic accident adds specificity to the study. The participant's demographic information and lack of history with prosthetic devices are clearly stated, contributing to the transparency and reproducibility of the research.

2.2. Data Acquisition: The section details the placement of Ag/AgCl foam electrodes on specific muscles of the participant's left side and the use of electronic boards for data acquisition. The description of the AD8232 integrated circuit and its adaptation for measuring muscle activity demonstrates the technical rigor of the study. However, further information on the calibration and validation of the equipment would enhance the reliability of the data collected.

2.3. The experimental setup, including the participant's posture and movement instructions, is well-described. The use of a visual interface to guide the participant and random selection of trials mitigates potential biases and ensures data integrity. The integration of an AI model to provide real-time feedback to the participant adds an innovative dimension to the study design.

2.4. Data Analysis: The section elucidates the process of data preprocessing, feature extraction, and model training using Edge Impulse and Python Keras. The rationale behind the selection of specific parameters and techniques is provided, enhancing the transparency of the analysis process. However, additional information on the validation and optimization of the machine learning models would strengthen the reliability and generalizability of the findings.

3. The Results section of the article "Embedded Machine Learning System for Muscle Patterns Detection in a Patient with Shoulder Disarticulation" provides a detailed account of the electronic board assembly, model training, testing, and online classification processes involved in the study.

3.1. The description of the construction of the multilayer copper-printed board based on the AD8232 integrated circuit design is comprehensive and well-illustrated. The implementation of a bandpass filter and standby mode for energy saving demonstrates thoughtful engineering considerations. The inclusion of graphs illustrating EMG signals from different muscle movements enhances the clarity of the presentation.

3.2. Model Training: The section elucidates the process of feature extraction using two distinct methods (Flatten and Spectral Analysis) and the subsequent training of machine learning models. The detailed explanation of parameter selection and model performance evaluation provides insights into the methodological rigor of the study. However, additional information on the validation and optimization of model architectures would enhance the reliability of the results.

3.3. Model Testing: The section presents the results of model testing on separate datasets, highlighting the accuracy and challenges encountered in predicting different muscle movements. The comparison between the Flatten and Flatten + Spectral Analysis models offers valuable insights into the relative effectiveness of feature extraction methods. However, further discussion on the implications of model performance discrepancies and potential strategies for improvement would enrich the analysis.

3.4. Online Classification: The section describes the real-time detection of muscle movements using the trained machine learning models. The presentation of accuracy results for the online classification process provides a comprehensive assessment of model generalizability. However, a more detailed discussion on the practical implications of the online classification results and their alignment with the study objectives would enhance the interpretation of the findings.

4. The discussion section provides a thorough analysis of the research findings, contextualizing them within the broader landscape of prosthetic control systems and highlighting their implications for future research and clinical applications.

Critical aspects of the Discussion section:

4.1. While the Discussion effectively summarizes the findings, it could benefit from a deeper interpretation of the results in the context of the study's objectives. Exploring the implications of specific findings on the broader field of prosthetic control systems and rehabilitation would add depth to the discussion.

4.2. Identifying and discussing the limitations and challenges encountered during the study would provide a more balanced perspective. Addressing factors such as the small participant pool, potential biases in data collection, and technical constraints of the prototype would enhance the transparency and credibility of the research.

4.3. While the Discussion compares the proposed system with existing solutions, a more critical analysis of the strengths and weaknesses of different approaches would enrich the discussion. Evaluating the limitations and potential biases of previous studies could provide valuable insights into the novelty and significance of the current research.

4.4. Assessing the generalizability of the findings beyond the specific experimental conditions is crucial. Discussing the extent to which the results can be extrapolated to other populations, types of amputations, and real-world settings would help contextualize the study's contributions and inform future research directions.

4.5. Considering the ethical and societal implications of the research is essential. Discussing factors such as privacy concerns, user autonomy, and the impact of technological advancements on healthcare delivery would broaden the scope of the discussion and highlight the broader implications of the study beyond technical aspects.

5. While this conclusion effectively summarizes the study's findings and implications, it suffers from being overly lengthy. The integration of AI into the portable EMG-based prosthetic control system presents promising advancements for upper limb amputees. However, the challenges in recognizing left arm movements should be addressed. The deployment of a low-power microcontroller platform underscores feasibility, but future research should broaden participant bases and explore diverse limb movements. The integration of AI models offers the potential for transformative prosthetic design. Despite its length, this conclusion marks a significant contribution to the field of prosthetic rehabilitation.

Reviewer 3 Report

Comments and Suggestions for Authors

The work is interesting for having investigated the use of the Edge Impulse platform with the related AI algorithms. The results are consistent with the possibility of controlling a prosthetic shoulder.

The main limitation of the study is that a patient equipped with a prosthetic shoulder must also have a prosthetic elbow, wrist and hand and therefore the classes to be detected with the AI algorithms are much higher than LEFT - UP-RIGHT. If we consider Flexion and Extension of the Elbow, pronation and supination of the Wrist, opening and closing of the hand, at least 6 other classes are added. Clearly just three electrodes are no longer enough but more are needed. For this reason, for proximal levels of amputation, the TMR (target muscle reinnervation) surgical technique is used to increase the sampling sites of EMG signals. Table 1 on line 265 reports: "Table 1. Electrical characteristics of commercial electromyographs and prototype devices used for rehabilitation and prosthetic control." The electrodes mentioned are NOT used for prosthetic control. The standard product for electrodes for prosthetic control is the OTTOBOCK electrode (https://shop.ottobock.us/c/Electrode/p/13E200~550).

The electrodes presented, despite having excellent characteristics, are difficult to be compatible with the prosthetic technique normally used.

I suggest a revision of the document focusing more on the opportunities of the Edge Impulse platform and possibly applying it to a patient with a level of trans-radial amputation where usually only the prosthetic hand is present.

Reviewer 4 Report

Comments and Suggestions for Authors

The manuscript, "Embedded Machine Learning System for Muscle Patterns Detection in a Patient with Shoulder Disarticulation," combines artificial intelligence methods with a portable EMG system for upper limb amputations. The study is interesting and could be applied in different Biomedical applications. However, some aspects must be addressed clearly.

Introduction: 

1- Additional references must be included, highlighting the most relevant ones.  

Materials and methods:

2. Why did this study only involve one participant? I believe that it would have been better to include more patients to assess the proposal methodology.  

3. Figure 1: Improve the scheme. It would be better to include a real picture of the system, PCB boards, or photos without showing the patient faces.

4. How many samples from 80 and 20 were used? 

5. Why was the Python Keras used instead of the Edge Impulse platform to apply IA? It is confusing; please explain it.

Results:

6. Please include more pictures of the electronic board connected to the shoulder volunteer.

7. Figure 5. What are the units and names of the vertical axis? Please include it.

8. Which methods from the impulse platform were implemented? The above is not clear. 

Comments on the Quality of English Language

Moderate editing of English language required

Round 2

Reviewer 1 Report

Comments and Suggestions for Authors

Authors answered the questions very well.

Reviewer 3 Report

Comments and Suggestions for Authors

Dear authors, I appreciated the effort made to answer my questions. I still have great doubts regarding the possibility of controlling a real prosthetic limb using the hand, wrist, elbow and shoulder with only 3 EMG signals. In my opinion, the proposed control of the shoulder alone is not sufficient for real benefit to the patient. My positive opinion on the publication is therefore aimed at inviting the authors to extend the study by trying to include further EMG signals that also allow other joints to be managed. Finally, I suggest you to include a prosthetist in your research team.

Reviewer 4 Report

Comments and Suggestions for Authors

The paper "Embedded Machine Learning System for Muscle Patterns Detection in a Patient with Shoulder Disarticulation"  has the potential and deserves to be published in the Sensors Journal as the remarks were addressed totally. 

Comments on the Quality of English Language

Minor editing of English language required